# Reliability of radiologists' first impression when interpreting a screening mammogram

Ziba Gandomkar[1]*, Somphone Siviengphanom[1], Mo'ayyad Suleiman[1], Dennis Wong[1], Warren Reed[1], Ernest U. Ekpo[1], Dong Xu[2], Sarah J. Lewis[1], Karla K. Evans[3], Jeremy M. Wolfe[4,5], Patrick C. Brennan[1]

1 Image Optimisation and Perception Group (MIOPeG), Discipline of Clinical Imaging, Faculty of Medicine and Health, University of Sydney, Sydney, NSW, Australia, 2 Department of Computer Science, The University of Hong Kong, Pokfulam, Hong Kong, 3 Department of Psychology, University of York, York, United Kingdom, 4 Brigham & Women's Hospital, Boston, MA, United States of America, 5 Harvard Medical School, Boston, MA, United States of America

* ziba.gandomkar@sydney.edu.au

**Data Availability Statement:** All relevant pooled data are within the paper. Moreover, all raw data related to readings can be found in the following

## Abstract

Previous studies showed that radiologists can detect the gist of an abnormality in a mammogram based on a half-second image presentation through global processing of screening mammograms. This study investigated the intra- and inter-observer reliability of the radiologists' initial impressions about the abnormality (or "gist signal"). It also examined if a subset of radiologists produced more reliable and accurate gist signals. Thirty-nine radiologists provided their initial impressions on two separate occasions, viewing each mammogram for half a second each time. The intra-class correlation (ICC) values showed poor to moderate intra-reader reliability. Only 13 radiologists had an ICC of 0.6 or above, which is considered the minimum standard for reliability, and only three radiologists had an ICC exceeding 0.7. The median value for the weighted Cohen's Kappa was 0.478 (interquartile range = 0.419–0.555). The Mann-Whitney U-test showed that the "Gist Experts", defined as those who outperformed others, had significantly higher ICC values (p = 0.002) and weighted Cohen's Kappa scores (p = 0.026). However, even for these experts, the intra-radiologist agreements were not strong, as an ICC of at least 0.75 indicates good reliability and the signal from none of the readers reached this level of reliability as determined by ICC values. The inter-reader reliability of the gist signal was poor, with an ICC score of 0.31 (CI = 0.26–0.37). The Fleiss Kappa score of 0.106 (CI = 0.105–0.106), indicating only slight inter-reader agreement, confirms the findings from the ICC analysis. The intra- and inter-reader reliability analysis showed that the radiologists' initial impressions are not reliable signals. In particular, the absence of an abnormal gist does not reliably signal a normal case, so radiologists should keep searching. This highlights the importance of "discovery scanning," or coarse screening to detect potential targets before ending the visual search.

## 1 Introduction

It has been established that radiologists can distinguish abnormal chest x-rays [1] or mammograms [2] from normal ones at an above-chance level based on less than a half-second image

public link: https://www.kaggle.com/datasets/zibaga/gist-of-the-abnormal-on-screening-mammograms.

**Funding:** PCB, JMW, SJL, KKE, DX, and ZG was funded by National Health and Medical Research Council (grant number APP1162872). Also, PCB, JMW, SJL, and KKE was funded by National Breast Cancer Foundation (IIRS-18-089). The funders had no role in study design, data collection and analysis, decision to publish, or preparation of the manuscript.

**Competing interests:** The authors have declared that no competing interests exist.

presentation. This observation led to the "holistic processing" model, which suggests radiologists' initial impressions about an image, or the gist signal, are being used to guide their gaze to suspicious areas. More recently, evidence shows that radiologists can identify the gist of the abnormal, in the normal breast contralateral to a cancer [3, 4] and prior normal mammograms for women, who were diagnosed with breast cancer at a later date [5–7], suggesting potential for breast cancer risk prediction [8]. The prior or contralateral images do not contain an overt sign of breast cancer. Therefore, the holistic processing model which suggests a localized source for the gist signal, cannot explain these new findings. However, they may best be explained by a two-pathway process. The two-pathway process suggests that both selective and nonselective pathways are utilized in the visual processing [9, 10]. The selective pathway combines features to recognize objects. It is called "selective" as a target object is specifically selected and processed for identification. The first type of the gist signal, which contains "where information" which guides the gaze to the object. The capacity to do this is limited by a one target at the time. The second pathway is the "non-selective" one, which extracts information about the entire visual field. This pathway helps us to rapidly categorize the image or image. In the case of mammographic image, it helps radiologists rapidly assess the image type (e.g. high risk or low risk) and does not support precise target identification (e.g., localization of the lesion) [9, 10].

The performance of radiologists varied considerably in gist experiments [11]. Although, inter-radiologist variations exist in usual viewing conditions [12, 13], with rapid detection of cancer the variation was significantly higher, with accuracy levels ranging from chance-level to performance levels comparable to usual reporting [11]. To utilize the signal rapidly extracted we refer to as the gist of the abnormal for improving breast cancer detection [14, 15] or identifying high-risk individuals [8], radiologists need to know whether they can trust their initial impression and whose gist signal is reliable. There might be a subset of "Gist Experts" whose first impressions are more accurate and reliable, even if that is not true of the larger radiologist population. Our work has shown that the abnormality rating in usual reporting and visualization conditions is not necessarily correlated with the radiologists' first impression about the case and the gist signal might be ruled out by the radiologists after a detailed inspection of the image [11]. Also, we showed that radiologists' overall performance, as measured by the area under the receiver operating characteristics curve (AUC), in the usual viewing condition is not correlated with their performance in the gist experiment, where their initial impression is based on a half-second image presentation was recorded [11]. In other words, the "Gist Experts", or those who outperformed other radiologists in the gist experiment, were not necessarily the "high-performers" in the usual viewing and reporting condition.

At this stage however, it is unknown whether the hypothesis of the existence of the "Gist Experts" can be supported even though for other visual perception tasks, which require holistic processing such as face recognition [16] or novel object recognition [17], individual differences in holistic processing were noted and participants' performances in two separate viewing rounds remained consistent [16]. None of the previous radiologic-based studies [3–8, 11, 18] explored if an observer's performances based on his/her initial impression on multiple occasions are associated. In other words, it is unclear whether an observer, who outperformed others in a single gist experiment, always performs better at detecting the gist of the abnormal or it is a random process, and no recommendation can be made for a specific observer about the accuracy of their initial impression. The test-reset reliability of the gist responses from an observer shows how strongly the gist responses from that observer in two rounds resemble each other. This is different from the consistency of overall performance, as for example similar sensitivity values in two rounds for detecting abnormal cases do not guarantee that an observer detected picked an exactly identical set of abnormal cases in two rounds.

It is also unknown whether the gist of the abnormal is present in a few specific abnormal cases and drives the above-chance performance in detecting abnormal cases or it is present generally across the abnormal set but picked up in a stochastic manner by the observer. Also, our earlier experiments showed that when we record the observers' initial impression on a scale a continuous scale, i.e., on a scale of 0 (confident normal) to 100 (confident abnormal), on some occasions an observer might detect a very strong gist signal in an image (a number near to 100), on some other occasions, an observer might indicate the gist of the abnormal is absent in an image (a number near to 0). It is unknown whether in these extreme occasions, an observer's initial impression can be trusted. The present study analyses such occasions.

In the present study, the gist responses from radiologists and breast physicians assessing screening mammograms were collected twice after a wash-out period of at least one month to explore:

1. if observer performances in two rounds are related (intra-radiologist variability in the overall performance).

2. if a subset of observers with higher performances or "Gist Experts" exists (i.e., a subset of readers who consistently outperformed others in the gist experiment),

   (2–1) to determine whether reader characteristics are associated with being a "Gist Expert" (exploring the association between the reader's performance and their characteristics).

   (2–2) to investigate the test-retest reliability of gist signals from "Gist Expert" and explore if they produce more reliable gist responses compared with "others".

3. whether the gist of the abnormal is present in a few specific abnormal cases and drives the above-chance performance in detecting abnormal cases or it is present generally across the abnormal set but picked up in a stochastic manner by the observer.

4. if readers should trust their initial impression when it strongly suggests the abnormality or normality of a case. To do so, the abnormality probability was explored in two extreme conditions: when an observer detects a very strong gist and when the gist of the abnormal was not (or poorly) perceived.

Aim (1) explores if the overall performances of a reader across all cases in two rounds (intra-radiologist variability in the performance) were associated, while aim (2–2) investigates how reliable rating is for a single case (test-retest reliability of gist signals). If the gist signal is generally present across the abnormal cases but picked up in a stochastic manner by an observer, the observer could detect the abnormality signal in one round but could miss it in another round. In such a scenario, an observer's overall performance could be almost similar in two rounds, however, each time they detect the gist of the abnormal in different cases. This translates into low intra-radiologist variability in the overall performance but poor test-retest reliability of the gist signal. While aims (1) and (2) focus on readers, aims (3) and (4) focus on the cases-level analysis to provide recommendations about whether an observer should trust their initial impression when a very strong gist of the abnormal is noted on a mammogram or when no signal was perceived.

## 2 Materials and methods

Ethics approval was obtained from the University of Sydney (2019/1017). All participants provided written informed consent prior to the data collection. The present study utilized the protocol, used in the previous studies to collect radiologists' initial impression, otherwise known as the gist response or gist signal. The protocol is explained below. In the experiment presented

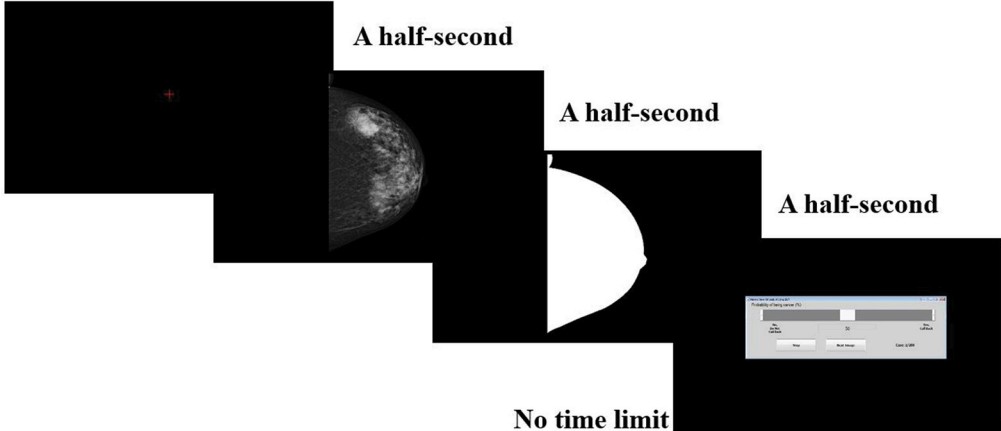

**Fig 1. The protocol for recording the rapid first impression signal (i.e., the gist of the abnormal).** A red cross was first presented for 500 milliseconds to ensure participants fixated at the display center. Then the mammogram was presented for 500 milliseconds. To stop image visual processing after 500 milliseconds, a white mask corresponding to the breast area was shown. Finally, the readers had unlimited time to indicate the probability of the case being abnormal on a scale of 0 (clearly normal) to 100 (clearly abnormal).

in the current study, gist responses from radiologists on an identical set of images were collected twice (i.e., Round 1 and Round 2), to explore the association between observers' performances in two rounds as well as the reliability of the gist signal. Using the data from this experiment, we also explored if reader characteristics are associated with the gist responses. Data collected from this experiment was also used for case-level analysis.

## 2.1 Protocol for recording the gist signal

To record the performance of rapid abnormality detection which we believe is partially based on detection of the gist signal, a multi-observer experimental protocol to assess for the presence of abnormality (Fig 1) was used [6]. As shown, a red cross symbol was first presented for 500 milliseconds to ensure participants fixated at the center of the display. This was then followed by presenting the mammogram for 500 milliseconds. To end the visual processing of the image, a mask corresponding to the breast area was then shown. Finally, the readers had unlimited time to indicate the probability of the case being abnormal on a scale of 0 to 100.

## 2.2 Sample size calculation

The details of the statistical analysis are described in Section 2–5. The main analysis, which required sample size calculation, was investigating the intra- and inter-reader reliability of the gist signal, as measured by the intraclass correlation coefficient (ICC). To do so, a priori sample size calculation was performed as suggested in [19]. For the intra-reader observer, where two assessments were available for each reader, an ICC of 0.6 (with an expected precision of 0.1 and a confidence interval of 95%) was assumed. An ICC of at least 0.6 is often used as the minimum standard for reliability coefficients in research [20]. Based on these assumptions, the minimum required sample size was 159. For the inter-reader reliability of the gist signal, we assumed assessments from at least 25 individuals per image were available. To detect an ICC of 0.6 with an expected precision of 0.1 and a confidence interval of 95%, the minimum required sample size was 50. Considering the calculated minimum sample sizes for investigating the intra- and inter-reader reliability, in our experiment, each participant was asked to

read 160 images. This sample size satisfied the minimum requirement for both purposes. Further details about the utilized image set are provided in Section 2–4.

## 2.3 Experimental design and study participants

A recruitment email was sent to 50 individuals, who previously attended BreastScreen Reader Assessment Strategy (BREAST) workshops [19] from 2017 to 2019 and indicated they are interested in participating in future research studies. All of these interpreted individuals read more than 2000 mammograms per year (based on both screening and diagnostic). In total, 39 readers agreed to participate. In the sample size calculation, the minimum number of participants is assumed to be 25. Therefore, the recruited number of readers was sufficient to detect an ICC of 0.6. To understand intra-reader variations in rapid cancer detection, each reader performed the readings twice with an identical set of images (i.e., Round 1 and Round 2). In each session, the images were shuffled and therefore, the order of presentation of images was randomly assigned to each round and each observer. Readers were alone in the room while assessing the images to ensure ratings were conducted independently. The washout period, i.e., the period between each experiment, was set to a minimum of 1 month to reduce recall bias. Readers have assessed the image based on a half-second image presentation. No further clinical information about cases was available to them. As each participant assessed the images twice, in total gist responses from 78 sessions were collected (resulting in 39 readers×2 rounds each = 78 sessions; each session contained scores on rapid first impression about image abnormality for 160 mammograms). Readers were also asked to complete a comprehensive questionnaire about their workload, practice type, experience levels, age, and gender. The questionnaire was designed based on earlier studies, exploring the association between reader characteristics and their performance in interpreting screening mammograms [20].

Prior to the experiment, the background information about the rapid sense of the gist of cancer and its two types (i.e., one with localized source and one related to the overall mammographic features) was explained to the observers. They were told that the purpose of this experiment is recording their first impression about the case based on the overall image characteristics. We did not provide a specific value for the prevalence of cancer in these cases. However, we did state that the prevalence of abnormal cases would be much higher than their usual practice. We did not mention the possibility that the images without overt signs of cancer could be priors from women who would develop cancer in the next round of screening (21 to 29 months later).

## 2.4 Mammograms

In each round, participants assessed 160 craniocaudal (CC) mammograms, presented in random order. These images were randomly retrieved from the breast cancer screening archive [19]. All images were acquired using full-field digital mammography and two experienced radiologists confirmed that the images had an acceptable quality in terms of breast positioning and acquisition parameters as per the criteria presented in [21]. Four categories, each containing 40 images were included: "Normal" mammograms came from women who were normal at the time of examination and remained normal in the next round of screening. The experienced radiologists were asked to ensure none of the "Normal" cases contained benign findings or regions suspicious of malignancy. "Cancer" images contained biopsy-proven malignancies. All "Cancer" cases were retrieved from an archive of BreastScreen Australia, where the standard practice is double reading with the arbitration. The selected cases were missed by one of the two radiologists who assessed the cases. The average tumor size was 10.6±5.2 and it ranged from 3mm to 26 mm with a median value of 10mm. The "Prior_Vis" and "Prior_Invis"

**Table 1. Characteristics of the mammograms.**

| Characteristic | Cancer | Prior_Vis | Prior_Invis | Normal |
|---|---|---|---|---|
| Density | | | | |
| BIRADS I (Fatty) | 4 | 8 | 5 | 8 |
| BIRADS II (Scattered fibroglandular) | 17 | 20 | 19 | 17 |
| BIRADS III (Heterogeneously dense) | 19 | 11 | 11 | 14 |
| BIRADS IV (Extremely dense) | 0 | 1 | 5 | 1 |
| Location of cancer sign | | | | |
| Central | 8 | 6 | 2 | - |
| Inner | 13 | 9 | 11 | - |
| Outer | 18 | 23 | 24 | - |
| Retro areolar | 1 | 2 | 3 | - |
| Lesion type | | | | |
| Architectural Distortion | 2 | 2 | 1 | - |
| Calcification | 5 | 6 | 8 | - |
| Discrete Mass | 6 | 11 | 4 | - |
| Non-specific density | 11 | 5 | 7 | - |
| Speculated Mass | 5 | 6 | 8 | - |
| Stellate | 11 | 10 | 12 | - |

categories were images with or without overt cancer signs, from women who were diagnosed at a later screen. The images were categorized in these groups based on the consensus of two experienced radiologists, who assessed images and pathology reports. The characteristics of cases in each one of the categories are presented in Table 1. As no cancer was present on Normal" images, the fields corresponding to the cancer characteristics are empty. For "Prior_Vis" cases the sign was visible but non-actionable. The values provided for "Prior_Vis" and "Prior_-Invis" categories were based on the subsequent screening examination, on which the cancer was detected. Our earlier studies showed that the gist of the abnormal can indicate the presence of current breast cancer as well as an elevated risk of future breast cancer [8]. In cancer-containing cases, both "what" and "where" information could lead to the gist response. In high-risk cases, which do not contain an overt cancer sign (i.e., "Prior_Invis" category), the "what" signal drives the gist of the abnormal, perceived by the observers. Our earlier studies showed that when the gist signal solely relies on "what" information, it is less strong [8]. Our previous study showed that by including 40 images in each of these four categories, we ensured the dataset covers a spectrum of images, on which the strength of the gist score (as recorded by the experimental protocol described in 2–1) ranges from low to high. Please refer to the S1 File (Analysis 1), where it is shown that the utilized image set resulted in a wide range of values for the gist scores.

## 2.5 Statistical analysis

For each image, two gist responses were available from each observer (Corresponding to Round 1 and Round 2). As stated, four categories of images were available in the dataset. For each observer in each round, the AUC values for discriminating "Cancer" from "Normal", "Prior_Vis" from "Normal", and "Prior_Invis" from "Normal" were calculated. To answer each study question (please refer to the Introduction section), the following steps have been taken:

*Aim (1)- Are observer performances in two rounds related?*

To answer this question, for each round the AUC values for three categorization tasks representing "Cancer" vs "Normal", "Prior_Vis" vs "Normal", and "Prior_Invis" vs "Normal" was calculated. To explore the association between the overall performance in two rounds, *Pearson's correlation* between the AUCs of two rounds were calculated for each categorization task.

*Aim (2–1)- Are reader characteristics associated with the reader's performance in the gist experiment?*

The Pearson correlation coefficient was used to explore the association between the reader's performance and the reader characteristics, which were collected as continuous/ordinal variables, including age, number of hours per week currently spent in reading mammograms, number of screening cases you read per week (screening interpretative volume), number of years certified as a BreastScreen reader, percentage of your time is dedicated to reading breast images, percentage of the time dedicated to reading the diagnostic mammograms, duration of fellowship training, number of years since fellowship training, number of years registered as a reader, number of years reading mammograms, number of breast biopsy examinations performed in the last 12 months, how often participate in a Multi-Disciplinary Meeting (MDT) in a month, and how often correlate/review radiology-pathology findings for biopsy cases in a month. For the dichotomous categorical variables (i.e., gender, whether they are radiologists or breast physicians, whether they are a screen reader, whether they are fellowship-trained, whether affiliated with a university or educational institutes, and whether they work full-time), Mann-Whitney U-test was used to investigate if the reader's performance varied across two categories.

The AUC value can be treated as a continuous variable showing the reader's overall performance. It can be also thresholded to produce a binary variable, showing those with superior capability of detecting the gist of the abnormal (i.e., "Gist Experts") and other radiologists. To cluster the readers as "Gist Experts" and "Others", AUC values for each reader were averaged and a mixture of Gaussian with two components was fitted to the AUC values. It should be noted that the average of all six AUC values (AUCs for discriminating "Cancer" from "Normal", "Prior_Vis" from "Normal", and "Prior_Invis" from "Normal" in two rounds) was considered. These AUC values were strongly correlated with the average AUC values of the first and second rounds (Pearson's correlation of 0.89 and 0.93, respectively). The threshold for categorization was set to $\frac{(\mu_1+\delta_1)+(\mu_2-\delta_2)}{2}$, where $\mu_1$ and $\mu_2$ are mean values of two components while $\delta_1$ and $\delta_2$ are standard deviations. After fitting the mixture model to the data, it turns out one-third of readers were categorized as "Gist Experts" using this approach. Once readers were clustered using their performance into "Gist Experts" and "Others" categories, the median value of each variable was reported for each continuous variable. The p-values from Mann-Whitney U test for continuous variable and z-test for two proportions were found to explore if reader characteristics significantly differed between the "Gist Experts" and "Others".

*Aim (2–2)- Is the gist signal reliable and do "Gist Experts" produce more reliable gist responses compared with "Others"?*

To measure the test-retest reliability of the gist signal, inter- and intra-observer variability of the gist signal was explored. To investigate the intra-observer variability, the ICC of the gist scores between the first and second rounds was calculated for each reader. We also calculated the weighted Cohen's kappa to measure the agreement between the two rounds. The ICC is a powerful reliability measure; however, as weighted Cohen's Kappa has been used to quantify inter- and intra-observer agreement in previous studies [12, 13], here we calculated both metrics. To calculate the weighted Cohen's Kappa, gist scores were discretised into six categories. For both ICC and weighted Cohen's Kappa analysis, the Mann-Whitney U-test was used to indicate if the intra-observer reliability is significantly higher among the "Gist Experts". Using

Pearson correlation for continuous/ordinal characteristics and Mann-Whitney U-test for the categorical variables, we explored if the reader characteristics were associated with the level of intra-observer agreement as measured by ICC and weighted Cohen's Kappa values.

To explore the inter-observer variability, the ICC, Weighted Kappa, and Fleiss Kappa were used. For calculating the ICC, the ratings from all observers in the first round were used. For calculating the Weighted Kappa values, we considered all possible pairs of radiologists to obtain a range for the Weighted Kappa. As another alternative, we also calculated the Fleiss Kappa, which measures the agreement level of a fixed number of classifying items. Unlike Cohen's kappa, which measures agreement between not more than two observers, Fleiss Kappa allows measuring agreement among more than two observers.

*Aim (3)- Is the gist of the abnormal present in a few specific abnormal cases or is it present generally across the abnormal set but picked up in a stochastic manner by the observer?*

To investigate if a few salient cases drive the detection of gist of the abnormality or case category was a predictor for the gist score and the gist of the abnormal is present across the set and identified by readers in a stochastic manner, we conducted two analyses. First, we omitted the top 10 cases in each one of the abnormal categories (25%). To do so, we calculated an average gist score per image, by averaging the abnormality scores given to that image across all 78 sessions. After omitting the 25% of "Cancer" cases with the highest average scores, the median for the AUC of "Cancer" vs "Normal" categorization was calculated to explore if the AUC values significantly dropped. We also investigated how many of the cases, with a high abnormality rating (i.e., cases in the upper quartile of the gist scores) were common across all 78 sessions.

*Aim (4)- Should readers trust their initial impression when it strongly suggests the abnormality or normality of a case?*

To answer this question two extreme conditions were considered: when a reader detects a very strong gist and when the gist of the abnormal was not (or poorly) perceived. We explored the probability that a case was abnormal when the gist rating was large compared to when it was small. In each session, 20 cases with the highest gist scores (High group; upper quintile) and 20 cases with the lowest gist scores (Low group; lower quintile) were considered. In both High and Low groups, we explored if the median count of "Cancers" significantly differed from that of "Normals" by using Mann-Whitney U-tests.

## 3 Results

### 3.1 The association between observers' performances in two rounds

Thirty-nine radiologists agreed to participate in the study, and all completed both rounds (no drop-out). On average, readers conducted the second round of the readings 7±1 weeks after the first round. Therefore, for each reader in each round, 160 gist scores corresponding to 160 images were available. The AUC values for discriminating "Cancer" from "Normal", "Prior_-Vis" from "Normal", and "Prior_Invis" from "Normal" were calculated for each observer and each round. The average AUC values for "Cancer" vs "Normal" categorization were 0.73 (CI: 0.69–0.77) and 0.72 (CI: 0.68–0.76) in two rounds. The AUC values of "Cancer" vs "Normal" categorization in two rounds ranged from 0.47 to 0.87 and differed significantly from chance-level in both rounds (p<0.001). The AUC values between the two rounds were significantly correlated with a Pearson's correlation coefficient of r = 0.62 (p<0.0001). For "Prior Vis" vs "Normal" and "Prior_Invis" vs "Normal" categorizations, AUC ranges were 0.41–0.74 and 0.38–0.66, respectively; both different from chance-level (p<0.001). The AUC for "Prior Vis" vs "Normal" categorizations, averaged at 0.60 (CI: 0.57–0.63) in the first round and averaged at 0.61 (CI: 0.58–0.65) in the second round. The Pearson's correlation coefficients between the AUC values of two rounds was 0.55 (p = 0.0003). For "Prior_Invis" vs "Normal"

**Table 2. The reader characteristics and their associations with the AUC values for Cancer-Both vs Normal- Both, Prior_Vis-Both vs Normal, and Prior_Invis- Both vs Normal.**

| Variables | Cancer-Both | Prior_Vis- Both | Prior_Invis- Both |
|---|---|---|---|
| Age | 0.06 | 0.14 | 0.04 |
| # Hours per week currently spent in reading mammograms | 0.28 | 0.12 | 0.13 |
| # Screening cases you read per week? | **0.33**[*] | 0.22 | 0.24 |
| # Years certified as a BreastScreen reader | 0.05 | 0.05 | 0.03 |
| Percentage of your time is dedicated to reading breast images | 0.13 | 0.01 | 0.12 |
| Percentage of the time dedicated to reading the diagnostic mammograms | 0.20 | 0.10 | 0.02 |
| Duration of fellowship training | 0.08 | -0.05 | -0.14 |
| # Years since fellowship training | 0.24 | 0.24 | 0.19 |
| # Years registered as a reader | 0.01 | -0.01 | 0.04 |
| # Years reading mammograms | -0.12 | -0.15 | 0.03 |
| # Breast biopsy examinations performed in the last 12 months | **0.32**[*] | 0.26 | 0.08 |
| How often participate in a Multi-Disciplinary Meeting (MDT) in a month | 0.17 | 0.13 | 0.06 |
| How often correlate/review radiology-pathology findings for biopsy cases in a month | 0.22 | 0.10 | 0.22 |

[*] Statistically-significant.

categorizations, the AUC values averaged at 0.54 (CI: 0.51–0.58) in the first round and averaged at 0.59 (0.56–0.62) in the second round. The Pearson's correlation coefficients between the AUC values of two rounds was 0.37 (p = 0.0196). Please refer to the S1 File (Supplementary Analysis 2) for the further analyses showing the association between observers' performances in two rounds.

## 3.2 The relationship between reader characteristics and their performances

The relationship between reader characteristics and AUC performances was measured using a Pearson correlation for continuous/ordinal characteristics and Mann-Whitney U-test for the categorical variables. Table 2 shows the Pearson's correlation coefficient for all continuous variables and the AUC values for Cancer-Both, Prior_Vis-Both, and Prior_Invis-Both (all versus Normal-Both). The analysis exhibited a significant but weak association with the personal characteristics of weekly interpretative workload and the number of breast biopsy examinations performed in the last 12 months. This association was observed only for the AUC of Cancer vs Normal. None of the categorical variables produced statistically significant effects (all p-values>0.05).

As stated, we also categorized readers as "Gist Experts" and "Others". Mann-Whitney U-test was used to explore if any of the reader characteristics differed significantly between the two groups. Table 3 shows that the percentage of time dedicated to the reading of the diagnostic mammograms differed significantly among "Gist Experts" and "Others", with no other differences being seen. In S1 File (Supplementary Analysis 2), we presented more detailed analysis, showing the differences between the "Gist Experts" and "Others" in their capabilities in perceiving different types of the gist information, i.e. "what" and "where" signals.

## 3.3 The intra- and inter-reader reliability of gist responses

Table 4 shows the ICC and Weighted Kappa of the gist scores between the first and second round. Considering a value of 0.60 as the minimum standards for reliability coefficients, only 13 radiologists had an ICC of ≥0.6 and the ICC values for only three radiologists exceeded 0.7. As shown in Table 4, both ICC and weighted Cohen's Kappa analysis showed that the intra-

**Table 3. The reader characteristics of "Gist Experts" and Others.**

| | All readers | Gist Experts | Others | p-values |
|---|---|---|---|---|
| # Hours per week currently spent in reading mammograms | | | | 0.783 |
| Median | 7.0 | 9.5 | 6.0 | |
| Min-Max | 1.0–32.0 | 1.0–30.0 | 1.0–32.0 | |
| 25th percentile-75th percentile | 4.0–14.3 | 3.0–19.0 | 4.0–12.0 | |
| # Screening cases you read per week? | | | | 0.336 |
| Median | 200 | 237 | 200 | |
| Min-Max | 5–800 | 5–800 | 8–450 | |
| 25th percentile-75th percentile | 141–300 | 150–400 | 110–287 | |
| # Years certified as a BreastScreen reader | | | | 0.351 |
| Median | 12 | 17.5 | 10 | |
| Min-Max | 0.5–30 | 0.5–22 | 0.5–30 | |
| 25th percentile-75th percentile | 4.2–20 | 7.5–20 | 3–20 | |
| Percentage of your time is dedicated to reading breast images (%) | | | | 0.169 |
| Median | 40 | 70 | 40 | |
| Min-Max | 2–100 | 5–95 | 2–100 | |
| 25th percentile-75th percentile | 13–79 | 45–78 | 11–76 | |
| Percentage of the time dedicated to reading the diagnostic mammograms (%) | | | | **0.004**[*] |
| Median | 6.3 | 24.6 | 2.5 | |
| Min-Max | 0–100 | 0–90.2 | 0–100 | |
| 25th percentile-75th percentile | 0–20.0 | 9.0–35.0 | 0–11.5 | |
| Duration of fellowship training (Months) | | | | 0.682 |
| Median | 6 | 6 | 6 | |
| Min-Max | 6–12 | 6–12 | 6–12 | |
| 25th percentile-75th percentile | 6–12 | 6–12 | 6–12 | |
| # Years since fellowship training | | | | 0.435 |
| Median | 7.5 | 10 | 7 | |
| Min-Max | 0.5–22 | 5–16 | 1–10 | |
| 25th percentile-75th percentile | 1–12 | 0–20.0 | 0–20.0 | |
| # Years registered as a screen reader | | | | 0.615 |
| Median | 20 | 21 | 18 | |
| Min-Max | 2–35 | 2–30 | 3–35 | |
| 25th percentile-75th percentile | 9–25 | 12–25 | 9–25 | |
| # Years reading mammograms | | | | 0.670 |
| Median | 21 | 21.5 | 21 | |
| Min-Max | 0–40 | 0–34 | 0–40 | |
| 25th percentile-75th percentile | 10–28 | 8.5–25 | 10.29 | |
| # Breast biopsy examinations performed in the last 12 months | | | | 0.278 |
| Median | 120 | 250 | 100 | |
| Min-Max | 0–1000 | 30–100 | 0–1000 | |
| 25th percentile-75th percentile | 52–287 | 77–400 | 50–250 | |
| How often participate in a Multi-Disciplinary Meeting (MDT) in a month | | | | 0.988 |
| Median | 3 | 3 | 3 | |
| Min-Max | 0–10 | 1–6 | 0–10 | |
| 25th percentile-75th percentile | 3–4 | 3–4 | 3–5 | |

*(Continued)*

**Table 3.** (Continued)

| | All readers | Gist Experts | Others | p-values |
|---|---|---|---|---|
| *How often correlate/review radiology-pathology findings for biopsy cases in a month* | | | | 0.330 |
| Median | 4 | 4 | 4 | |
| Min-Max | 0–100 | 1–100 | 0–60 | |
| 25th percentile-75th percentile | 3–15 | 4–40 | 3–10 | |
| *P-values from z-test for two proportions; Count (Number) are presented* | | | | |
| Gender (Female or Male) | 24 (Female) | 10 | 14 | 0.061 |
| Discipline is (Radiologist or Breast physician) | 36 (Radiologist) | 11 | 25 | 0.920 |
| Being a screen reader (Yes/No) | 5 (No) | 1 | 4 | 0.575 |
| Fellowship-trained (Yes/No) | 14 (Yes) | 5 | 9 | 0.617 |
| Whether affiliated with a university or educational institutes (Yes/No) | 21 (Yes) | 7 | 14 | 0.711 |
| Working full-time (Yes/No) | 21 (Yes) | 4 | 17 | 0.087 |

The first section of the table shows the continuous variables while the second section shows the categorical ones. For continuous variables, the median values and p-values from Mann-Whitney U test are shown while for the categorical ones, number of readers in each category and the p-values from z-test for two proportions are shown.

\* represents significant p-values.

**Table 4. The inter-class correlation coefficient (ICC) and Weighted Kappa values for measuring the intra-observer variability in Round 1 and Round 2 and the ICC, Weighted Kappa, and Fleiss Kappa for measuring the inter-observer variability of the gist signal.** For inter-reader ICC value and Fleiss Kappa, the 95% confidence interval (CI) is also reported.

| | Metric | All | Gist Experts | Others |
|---|---|---|---|---|
| Intra-observer | ICC | 0.519 | 0.628 | 0.503 |
| | Median of all readers* | | | |
| | Min value | 0.347 | 0.474 | 0.347 |
| | Max value | 0.756 | 0.756 | 0.724 |
| | 25th percentile | 0.467 | 0.544 | 0.415 |
| | 75th percentile | 0.623 | 0.687 | 0.580 |
| | Weighted Kappa | 0.478 | 0.544 | 0.467 |
| | Median of all readers** | | | |
| | Min value | 0.295 | 0.396 | 0.295 |
| | Max value | 0.708 | 0.698 | 0.708 |
| | 25th percentile | 0.419 | 0.479 | 0.404 |
| | 75th percentile | 0.555 | 0.652 | 0.517 |
| Inter-observer | Weighted Kappa | | | |
| | Median of all readers | 0.3490 | 0.3215 | 0.360 |
| | Min value | -0.0243 | -0.0243 | 0.0752 |
| | Max value | 0.6322 | 0.6042 | 0.6322 |
| | 25th percentile | 0.2435 | 0.1776 | 0.2762 |
| | 75th percentile | 0.4326 | 0.4162 | 0.4381 |
| | ICC | 0.306 [95% CI: 0.26, 0.36] | 0.321 [95% CI: 0.26, 0.39] | 0.299 [95% CI: 0.25, 0.36] |
| | Fleiss Kappa | 0.106 [95% CI: 0.105, 0.106] | 0.093 [95% CI: 0.091, 0.095] | 0.108 [95% CI: 0.107, 0.109] |

\* p-value = 0.0026

\*\* p-value = 0.0263

observer reliability is significantly higher among the "Gist Experts". The Mann-Whitney U-test showed that the "Gist Experts" had significantly higher ICC values (p = 0.002) and Kappa values (p = 0.026). Therefore, the "Gist Experts" had higher ICC and Kappa values, and hence produced more reliable gist scores. As per definition a weighted Cohen's Kappa of 0.80 or above and an ICC of at least 0.90 exhibit strong agreement level. Therefore, based on both metrics, even for the "Gist Experts" within the fourth quartile (those readers with an ICC between 75th percentile and max in Table 4), the intra-radiologist agreements were not strong. None of the reader characteristics showed a statistically significant (all p-values>0.05) association with the intra-radiologists ICC and weighted Cohen's Kappa values.

The gist signal showed poor inter-reader reliability (ICC: 0.31, CI: 0.26–0.37). We also considered all possible pairs of radiologists in Round 1 and calculated the Weighted Kappa. It ranged from -0.02 to 0.63 representing no to moderate agreement. The median value was 0.35, showing only fair level of agreement. The Fleiss Kappa of 0.11 (CI: 0.105, 0.106), which exhibits slight agreement, resonates with the findings from the ICC and Weighted Kappa analysis.

### 3.4 Case-level analysis

After omitting the 25% of "Cancer" cases with the highest average scores, the median for the AUC of "Cancer" vs "Normal" categorization dropped from 0.74 to 0.70 but still differed significantly from the chance-level (p<0.001). After omitting the 25% of "Prior_Vis" cases with the highest average scores, for "Prior_Vis" vs "Normal" the median AUC values dropped from 0.62 to 0.57 (significantly better than chance-level p<0.001) while the median AUC for "Prior_Invis" vs "Normal" dropped from 0.55 to 0.50 (p = 0.10) after omitting 25% of "Prior_Invis" cases.

To further explore if the significant p-values for the AUCs are due to a few specific cases, we explored how many of the cases, with a high abnormality rating (i.e., cases in the upper quartile of the gist scores) were common across all sessions. Only four "Cancer" cases were in the top quartile of more than 75% of sessions. There were 18 "Cancer" cases, 4 "Prior_Vis" cases, and 2 "Prior_Invis" cases, which were in the top gist quartile of more than 50% of the sessions. These 24 cases constitute 40.6% of the datapoints, i.e., 1266 from 40 (number of cases in the upper quartile of a session) × 78 (number of sessions). When we excluded these 24 cases, all three AUCs remained significantly different from the chance-level (p<0.001). Therefore, although certain cancer cases had, on average, a stronger signal of cancer based on rapid gist detection and the likelihood that a reader detects the gist of abnormal on those cases was higher, detecting gist is a stochastic process and an observer may or may not perceive it in a session. To further explore if the gist of abnormal the radiologists perceive is an unvarying image property, the ratings of likelihood abnormality given by all 39 readers to each case in a round were averaged. Such averaging cancels out the noise in the gist responses. The ICC value comparing for these average abnormality ratings between two rounds was 0.96 (CI: 0.95–0.97).

### 3.5 Abnormality and normality likelihood in the case of strong or weak gist signal

Fig 2 shows the category membership of the 20 cases with the highest rapid abnormality ratings in each session. Each row represents one session for one reader. Categories are color-coded. As one would hope, in general, when the rating of gist abnormality is high, cases are not normal. If cases were randomly selected, 25% of cases would have been selected from each category. Here, of the 20 cases, an average of 11±2 cases were from the Cancer category while

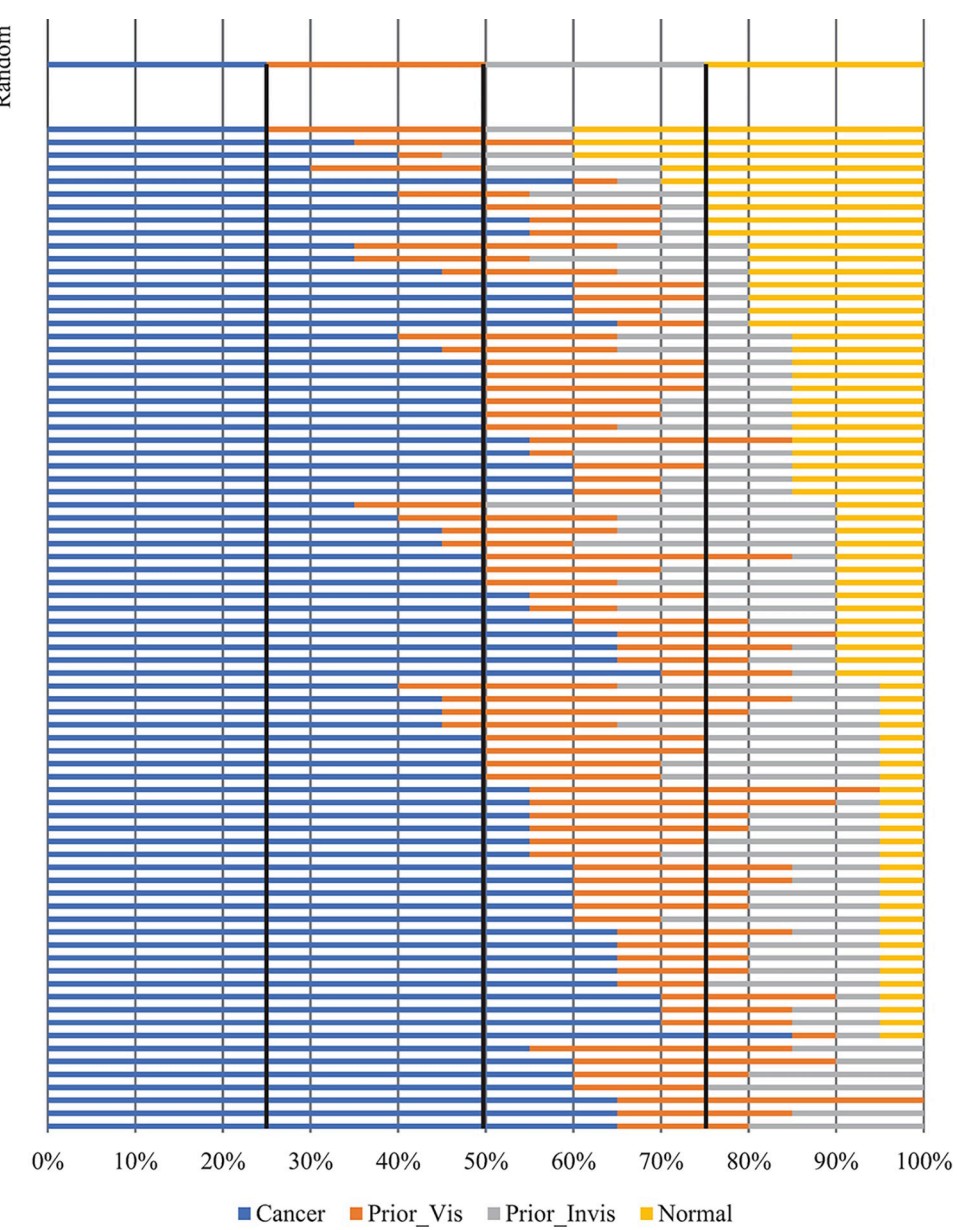

**Fig 2. Distribution of 20 cases with the highest gist of abnormality ratings in each session across various categories.** In total, 78 sessions (two rounds from 39 readers) were available. The random level is shown by the black lines.

just 2±1 cases were normal (p<0.001). Thus, if the gist rating is high, odds are that the case is abnormal.

If we look at the 20 cases with the lowest scores on the abnormal gist rating an average of 5 ±3 cases were from the Cancer category while 4±3 cases were normal. The difference was non-significant (p = 0.072), suggesting that a low score did not constitute convincing evidence that a case was normal. Fig 3 shows the category membership of the 20 cases with the lowest gist response.

As implied by Fig 2, among the 20 cases with the highest gist scores, counts of the "Cancers" were higher than counts of the "Prior_Vis", which were in turn higher than counts of the

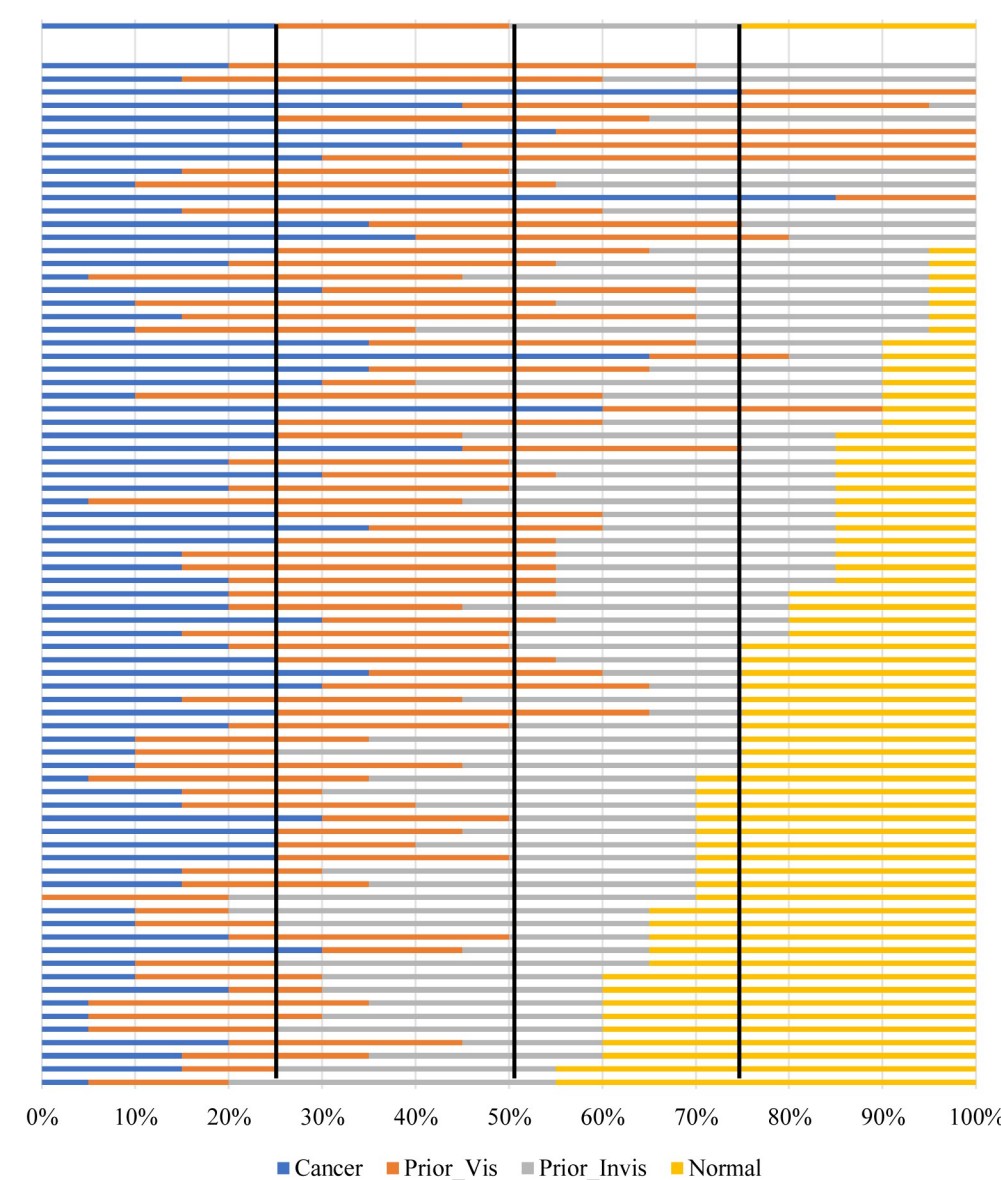

**Fig 3. Distribution of 20 cases with the lowest gist of abnormality ratings in each session across various categories.** In total, 78 sessions (two rounds from 39 readers) were available. The random level is shown by the black lines.

"Prior_Invis" and "Normals". To explore the trend, the boxplots showing the distribution of counts in each image category were created and shown in Fig 4. The boxplot for each category shows the distribution of counts Similar boxplots and a trendline were also shown for the 20 cases with the lowest gist scores. As shown, in the absence of the gist of the abnormal, no trend was observed for the counts of different image categories.

## 4 Discussion

Earlier studies showed the accuracy of radiologists in detecting breast cancer, when relying only on their initial impression about a screening mammogram ranges from chance-level to performance levels comparable to usual reporting [11]. However, none of these studies have investigated the test-retest reliability of the gist responses and if certain observers consistently

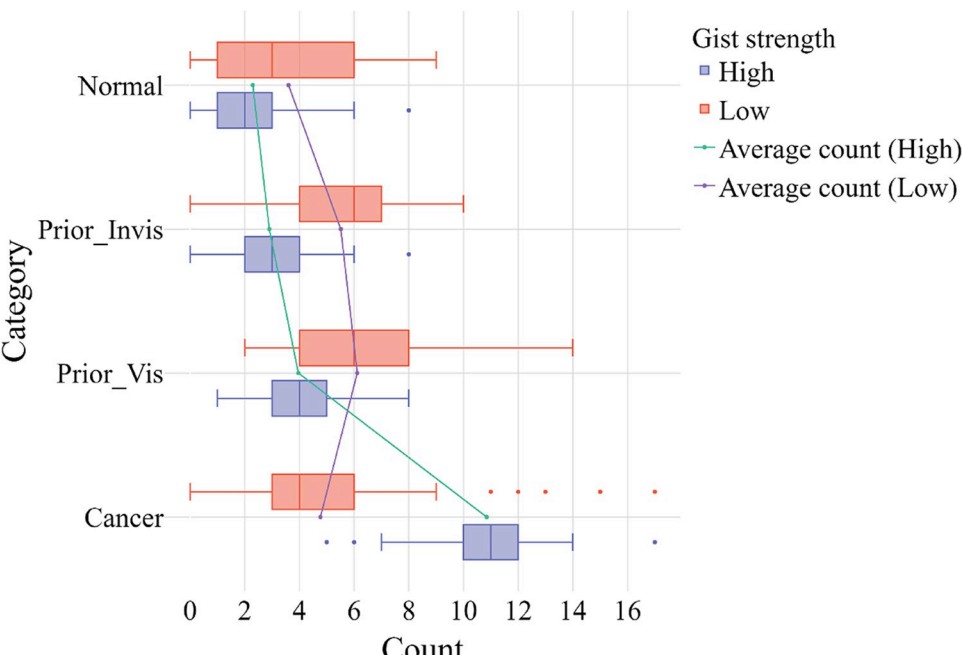

**Fig 4. In each trial, 20 cases with the highest (High) and lowest (Low) gist scores were considered and count of these cases in each category were found.** The boxplots show the distribution of counts across various categories. The average counts (showing the average trial) are also plotted. As shown, for case in High gist group, a trend from Cancer to Normal is evident while such trend is absent in Low group.

perform better at detecting the gist of abnormalities. As a result, it is unknown if high-performing observers in one gist experiment can reliably trust their initial impression about the presence of an abnormality in new cases. Our data indicated that the radiological initial impression or gist signal is noisy with low test-retest reliability.

Both ICC and Cohen's kappa values suggested poor to modest intra-reader agreement for the gist signal. Therefore, unlike usual reporting, where radiologists exhibit strong level of agreement with themselves [13, 22–25], in gist experiments, intra-observer reliability of the signal is poor. The analysis of the readers' overall performance (AUC analysis) showed that when the gist signal was strong, i.e., in cases of cancer, the overall performance level in the first round of the gist experiment remained relatively unchanged in the second round. In spite of that, the low ICC and weighted Cohen's kappa values suggest that the gist signal picked up in a stochastic manner by an observer in each round and cases led to a strong signal in the first round, did not necessarily result in a strong signal in the second round.

We used the AUC value, showing the reader's overall performance, to categorise readers as "Gist Experts" and "other radiologists". "Gist Experts" were readers with superior capability of detecting the gist of the abnormal. Statistical analysis showed that both ICC and weighted Cohen's Kappa values were significantly higher among the "Gist Experts". Hence, gist signal from the "Gist Experts" had higher level of intra-observer reliability. However, even for the experts, the ICC values for all readers were below 0.75 and experts exhibited poor to moderate level of test-retest reliability. In intra-observer reliability analysis, usually an ICC of 0.90 or above is considered as excellent reliability and an ICC between 0.75 and 0.9 indicates good reliability [26]. Therefore, none of our readers exhibited an excellent or even good level of intra-observer reliability and the gist signals from majority of the experts (about 75%) had only a moderate reliability.

The inter-reader reliability of the gist signal was poor as measured by both ICC (0.31, CI = 0.26–0.37). and Fleiss Kappa (0.106, CI = 0.105–0.106). Even when the inter-reader reproducibility was analyzed within the "Gist Expert" group, the signal was not reliable enough to be used in clinical practice. Compared to usual viewing conditions [13, 22–25], the magnitude of inter-reader variability is larger in the gist experiment. Considering the noisiness of detecting the gist of the abnormal and much less information given to the readers about the case and also the availability of other information about the case while interpreting cases in standard viewing, a lower magnitude of intra- and inter-observer agreement, compared to the usual viewing condition, was expected.

None of the earlier gist studies conducted the case-level analysis to investigate if an observer's initial impression can be trusted in extreme occasions, where a very strong or poor gist signal was detected. Our findings suggest that a strong gist of the abnormal, should not be ignored, because when the gist is strong, the probability of a case being abnormal is significantly higher than the probability that it is normal. Conversely, in the absence of a sense of an abnormal gist, readers should keep searching because the absence does not reliably signal a normal case. This finding provides evidence supporting the importance of "discovery scanning", which refers to coarse screening of the mammogram to detect potential target features before terminating the visual search [27]. It is usually recommended to actively search for alternate hypotheses in the image interpretation to avoid anchoring bias where radiologists would remain fixed at their initial diagnosis and actively search for evidence to confirm their initial hypothesis [28, 29]. Our data shows, particularly, when radiologists' first impression is normal (absence of the gist of abnormal), they should be mindful of anchoring bias [28, 29] and should not solely trust their initial impression.

Our study had a few limitations. First, although the intra- and inter-radiologist variabilities in the usual reporting conditions were compared to the existing literature, the exact range for our participants is not established. Moreover, the prevalence of abnormal and high-risk women was higher than the real clinical practice and the laboratory effect [30] could limit the generalisability of our findings. In addition, the recruited readers in the current study are mostly screen readers, even readers who were categorized as "Others" and were not Gist Experts had an average annual screening interpretive volume of 10,000+ cases. This could limit the generalisability of the presented results for readers with low interpretive volume. Also, it should be acknowledged that a study with more readers could yield a significant difference for some of the reader characteristics. Moreover, we relied on self-reported measures for describing reader characteristics. Although for some of the measures such as whether they did a fellowship or not or their age, self-reported values should be accurate, measures such as weekly reading volume or diagnostic focus should be validated in the future as concerns about a possible overestimation of these measures have been raised previously [31]. Finally, in this study, we did not include cases with benign findings in our "Normal" category. Moreover, we did not include the normal cases highly suspicious of malignancy. We only focused on two extremes, i.e., normal and abnormal, to study the gist signal. These two categories resulted in low and high gist scores. By including "Prior-Vis" and "Prior-Invis", we ensured our test set also included images with intermediate gist scores to cover a range of values for the gist responses (Please refer to Analysis 1 in the S1 File). However, in real clinical practice, radiologists deal with benign findings and highly suspicious normal cases as well. As no benign cases were included here, we cannot exclude that a benign finding might evoke a gist signal as well. This could limit the usefulness of the gist signal for breast cancer detection in real clinical practice. In terms of the applicability of the gist signal to breast cancer prediction, future experiments with various types of benign cases should be conducted. Previous findings suggest that while proliferative benign diseases were associated with an increased breast cancer risk, non-

proliferative benign diseases do not increase the risk of a future malignancy [32]. Therefore, it should be tested whether the gist of the abnormal is stronger for cases with proliferative benign compared to the ones with non-proliferative benign diseases. Moreover, the histopathological characteristics of the included cases was not available. Some of the cancer types are more aggressive than other ones [33], hence, another potential future work could be investigating if the strength of the gist signal is related to the histopathological features of a lesion.

In summary, while previous work established the existence of a gist signal associated with the presence and even with the future likelihood of cancer, the current study shows that detecting this signal is noisy, with a relatively low level of inter- and intra-reader agreement. Therefore, radiologists should be mindful of bias from their initial impression about the case and the gist signal cannot be reliably used in clinical practice.

## Supporting information

**S1 File.**
(DOCX)

## Author Contributions

**Conceptualization:** Ziba Gandomkar, Mo'ayyad Suleiman, Warren Reed, Ernest U. Ekpo, Dong Xu, Sarah J. Lewis, Karla K. Evans, Jeremy M. Wolfe, Patrick C. Brennan.

**Data curation:** Ziba Gandomkar, Somphone Siviengphanom, Mo'ayyad Suleiman, Dennis Wong.

**Formal analysis:** Ziba Gandomkar, Dennis Wong, Sarah J. Lewis.

**Funding acquisition:** Ziba Gandomkar, Ernest U. Ekpo, Dong Xu, Karla K. Evans, Jeremy M. Wolfe, Patrick C. Brennan.

**Investigation:** Ziba Gandomkar.

**Methodology:** Ziba Gandomkar, Warren Reed, Ernest U. Ekpo, Dong Xu, Sarah J. Lewis, Karla K. Evans, Jeremy M. Wolfe, Patrick C. Brennan.

**Project administration:** Ziba Gandomkar, Somphone Siviengphanom, Mo'ayyad Suleiman, Ernest U. Ekpo, Sarah J. Lewis, Patrick C. Brennan.

**Resources:** Ziba Gandomkar, Warren Reed, Ernest U. Ekpo, Patrick C. Brennan.

**Software:** Ziba Gandomkar, Somphone Siviengphanom.

**Supervision:** Warren Reed, Sarah J. Lewis, Karla K. Evans, Patrick C. Brennan.

**Validation:** Ziba Gandomkar, Dennis Wong, Jeremy M. Wolfe, Patrick C. Brennan.

**Visualization:** Ziba Gandomkar, Sarah J. Lewis.

**Writing – original draft:** Ziba Gandomkar.

**Writing – review & editing:** Somphone Siviengphanom, Mo'ayyad Suleiman, Dennis Wong, Warren Reed, Ernest U. Ekpo, Dong Xu, Sarah J. Lewis, Karla K. Evans, Jeremy M. Wolfe, Patrick C. Brennan.

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
