## [Decision Letter · Decision Letter 0]

31 Jan 2022

PONE-D-21-31110Reliability of radiologists’ first impression when interpreting a screening mammogramPLOS ONE

Dear Dr. Ganodmkar:

Thank you for submitting your manuscript to PLOS ONE. After careful consideration, we feel that it has merit but does not fully meet PLOS ONE’s publication criteria as it currently stands. Therefore, we invite you to submit a revised version of the manuscript that addresses the points raised during the review process.

 This application has comments for major and minor revisions as noted by the reviewers.  Please address the concerns in a revised application.

We look forward to receiving your revised manuscript.

Kind regards,

Gayle E. Woloschak, PhD

Academic Editor

PLOS ONE

Journal Requirements:

PCB, JMW, SJL, KKE, DX, and ZG was funded by National Health and Medical Research Council (grant number APP1162872).

PCB, JMW, SJL, and KKE was funded by National Breast Cancer Foundation (IIRS-18-089).

3. Thank you for stating the following in the Acknowledgments/ Funding Section of your manuscript: 

This study was funded by National Health and Medical Research Council (grant number APP1162872) and National Breast Cancer Foundation (IIRS-18-089).

PCB, JMW, SJL, KKE, DX, and ZG was funded by National Health and Medical Research Council (grant number APP1162872).

PCB, JMW, SJL, and KKE was funded by National Breast Cancer Foundation (IIRS-18-089).

NO authors have competing interests

Additional Editor Comments:

One reviewer recommended major revision, another recommended minor revision. Please address all comments in the revised manuscript.

Reviewers' comments:

Reviewer's Responses to Questions

**Comments to the Author**

1. Is the manuscript technically sound, and do the data support the conclusions?

Reviewer #1: Partly

Reviewer #2: Partly

2. Has the statistical analysis been performed appropriately and rigorously? 

Reviewer #1: Yes

Reviewer #2: I Don't Know

3. Have the authors made all data underlying the findings in their manuscript fully available?

Reviewer #1: No

Reviewer #2: No

4. Is the manuscript presented in an intelligible fashion and written in standard English?

Reviewer #1: Yes

Reviewer #2: Yes

5. Review Comments to the Author

Reviewer #1: aaaaaaaaaaaaaaaaaaaaaaaaaaaaaaaaaaaaaaaaaaaaaaaaaaaaaaaaaaaaaaaaaaaaaaaaaaaaaaaaaaaaaaaaaaaaaaaaaaaaaaaaaaaaaaaaaaaaaaaaaaaaaaaa

aaaaaaaaaaaaaaaaaaaaaaaaaaaaaaaaaaaaaaaaaaaaaaaaaaaaaaaaaaaaaaaaaaaaaaaaaaaaaaaaaaaaaaaaaaaaaaaaaaaaaaaaaaaaaaaaaaaaaaaaaaaaaaaa

Reviewer #2: Dear Authors

This is an interesting and ambitious study exploring the gist signal in breast imaging reading by radiologists. A large number of readers were involved in the experiment on a sufficient number of cases. Even if the gist signal does not improve overall performance, the study finds, among other findings, that the gist signal is reliable in gist experts and that the signal should be trusted if the gists suggest a high likelihood of malignancy, but should be followed by a scrutinized search at low-likelihood. Those that spend more time in a diagnostic setting are more likely gist experts.

I have some concerns, mainly of clinical character, that could be clarified by some additions.

A broaden description of the cases included in the experiments are warranted (Materials and Methods/Mammograms). Did normal cases include benign findings or no findings at all? If no findings, we cannot exclude that also benign findings evoke a gist signal with the same magnitude as cancer cases. The gist signal might then not be as “useful” in a clinical setting where there are many benign findings. I can understand that you pick out the extremes (normal-abnormal) to study the gist signal, but this limitation should then be included in the discussion.

You state that all cancer cases were subtle. What is your definition of a subtle cancer case? Do you have some numbers to support the case selection, for example tumor size? The case illustrated in Fig 1. does not look like a typically subtle case.

Since breast cancer is a heterogeneous disease with different mammographic appearances, and therefore more or less difficult to detect, it could also be relevant to include tumor types. I wonder for example if invasive lobular cancer or microcalcification cases were included?

Concerning the readers. Could you check the validity that the readers on average read 200 cases per week/10,000 per year working 40% with breast imaging? It raises some concerns on the self-reported statistics (I am not implying that the readers are lying, but it can perhaps be hard to estimate your own work).

How come the time dedicated to reading diagnostic mammograms (diagnostic focus) was associated with gist experts in Table 2 when it was an unsignificant factor in Table 3, or do I read it wrong?

A few minor comments:

Abstract:

The labels Chigh and Clow do not appear later in the text. (Why C? It refers to high and low gist signal, right?)

Materials and Methods:

2-2 Mammograms. Explain CC.

Results:

Check “Prio_Vis” for consistency (incl. Discussion).

6. PLOS authors have the option to publish the peer review history of their article (what does this mean?). If published, this will include your full peer review and any attached files.

Reviewer #1: No

Reviewer #2: No

---

## [Author Response · Author response to Decision Letter 0]

4 Apr 2022

Please see the attached point-by-point responses to the reviewer's comments as I could not paste the figures here.

We would like to thank the editor and the reviewers for their valuable comments, which we believe have greatly improved the manuscript. Responses to specific comments made can be found below. The changes are also highlighted in the manuscript. 

 Also, all de-identified collected data from radiologists, used in this study will be available through the following Kaggle dataset link:

https://www.kaggle.com/datasets/zibaga/gist-of-the-abnormal-on-screening-mammograms

PCB, JMW, SJL, KKE, DX, and ZG was funded by National Health and Medical Research Council (grant number APP1162872). Also, PCB, JMW, SJL, and KKE was funded by National Breast Cancer Foundation (IIRS-18-089). The funders had no role in study design, data collection and analysis, decision to publish, or preparation of the manuscript. We can confirm that we understand the above-mentioned statement will be included as "Funding Statement" in the manuscript. 

The authors have declared that no competing interests exist.

Point by point responses to the comments:

Reviewer #2: 

1) I have some concerns, mainly of clinical character, that could be clarified by some additions. A broaden description of the cases included in the experiments are warranted (Materials and Methods/Mammograms). Did normal cases include benign findings or no findings at all? If no findings, we cannot exclude that also benign findings evoke a gist signal with the same magnitude as cancer cases. The gist signal might then not be as “useful” in a clinical setting where there are many benign findings. I can understand that you pick out the extremes (normal-abnormal) to study the gist signal, but this limitation should then be included in the discussion.

Thank you very much for the comment. To address this comment, we added this point as a limitation of our study and acknowledged that we only focused on two extremes (normal-abnormal) in this paper to study the gist signal. 

The abnormal categories were “Cancer”, “Prior_Vis”, and “Prior_Invis”. Certain types of benign findings are associated with an elevated risk of future breast cancer. As stated, two types of gist signal exist, one providing “where” information (to guide the gaze to suspicious areas) and one providing “what” information (to globally assess the image and alert the radiologist that the mammogram represents a high-risk patient). Therefore, the latter type of the gist signal could be present in these benign cases. If the purpose of using the gist signal is the prediction of a future malignancy, the existence of the gist of the abnormal on these benign cases could be desirable. However, further studies are required to compare the strength of the gist signals on mammograms from patients with non-proliferative benign breast diseases and proliferative benign breast diseases. As existing literature suggests that only cases with proliferative benign breast diseases demonstrated an increased risk of a future malignancy [1]. Therefore, in future, it should be explored if the gist signal is stronger in these cases compared to cases with non-proliferative benign breast diseases.

References 

1. Dyrstad SW, Yan Y, Fowler AM, Colditz GA. Breast cancer risk associated with benign breast disease: systematic review and meta-analysis. Breast Cancer Res Treat. 2015;149(3):569-75. Epub 2015/02/01. doi: 10.1007/s10549-014-3254-6. PubMed PMID: 25636589.

2) You state that all cancer cases were subtle. What is your definition of a subtle cancer case? Do you have some numbers to support the case selection, for example tumor size? The case illustrated in Fig 1. does not look like a typically subtle case.

Thank you for the comment. We deleted the word “subtle” and included the tumor size of the “Cancer” cases. All cases were retrieved from an archive of BreastScreen Australia, where the standard practice is double reading. The selected “Cancer” cases were missed by one of the two radiologists who assessed the cases. We added this to the manuscript instead of using the word “subtle”. 

3) Since breast cancer is a heterogeneous disease with different mammographic appearances, and therefore more or less difficult to detect, it could also be relevant to include tumor types. I wonder for example if invasive lobular cancer or microcalcification cases were included?

Thank for the comment. To address the comment, we added a table and provided the characteristics of lesions and also the breast density of the cases. However, the histopathological reports of the cases were not available. We acknowledged this as a limitation of our study in the discussion. This is an interesting future work, as the aggressiveness of various cancer types differs, and this could impact the gist strength.

4) Concerning the readers. Could you check the validity that the readers on average read 200 cases per week/10,000 per year working 40% with breast imaging? It raises some concerns on the self-reported statistics (I am not implying that the readers are lying, but it can perhaps be hard to estimate your own work).

Thank you very much for raising this interesting point. We added a few sentences in the discussion and explained that we relied on self-reported metrics and these metrics could be over-estimated. As the reviewer suggested, previous studies also raised concerns about the accuracy of self-reported reader characteristics. 

One possible explanation for these numbers is that there is a well-known problem of radiology workforce shortage in Australia and New Zealand [1-3]. Therefore, radiologists might work very long hours on their working days. Actually, on average radiologists in Australia and New Zealand work more than one full-time employee (FTE) [4]. 

In terms of the possibility of reading 200 cases per week, based on our own experiments at BreastScreen Reader Assessment Strategy (BREAST) workshops, on average, it took two hours for a screen reader to interpret a test of 60 cases. This equates to approximately 7 hours for interpreting 200 cases. Based on another study, on average, it took 240 seconds to interpret a digital mammogram [5]. Considering this figure, reading 200 cases equates to 13 hours. This study was conducted more than a decade ago, so considering the technological advancements, currently, the required time could be even lower. Considering this number, it may take 2 days to read 200 images. With workforce productivity of 75%, this equates to 40% of a working week if the individual works full-time. 

In Australia and New Zealand, there are about 2000 radiologists, and based on a self-reported survey about 14% of these radiologists are screen readers (in our cohort only 5 were not screen readers). In total, in each round of screening (i.e., 2 years), about 2.2M screening exams are produced. Because the standard practice is double reading with the arbitration, more than 2.2M assessments are conducted each year. Therefore, the average reading volume of screen readers is more than 7857 (2.2M assessments / (2000*0.14 screen readers). This equates to 164 cases per week (a person has 48 working weeks in a year in Australia and New Zealand). Therefore, even though it is close to 200, as raised by the reviewer, there is a probability that the respondents overestimated their reading volume. This is coherent with concerns raised by earlier studies [6]. 

References

1. Brooks PM, Lapsley HM, Butt DB. Medical workforce issues in Australia:" tomorrow's doctors-too few, too far". Medical Journal of Australia 2003;179:206-208.

2. Patty A. Radiologist shortage for breast scans. Daily Telegraph 2003.

3. Association AM. AMA blames Qld Health for radiologist shortage. ABC Premium News, Sydney 2006.

4. RANZCR, 2016 Australian radiology workforce report. Available from https://www.ranzcr.com/documents/4624-2016-clinical-radiology-workforce-census-report-australia/file

5. Haygood TM, Wang J, Atkinson EN, Lane D, Stephens TW, Patel P, Whitman GJ. Timed efficiency of interpretation of digital and film-screen screening mammograms. AJR. American journal of roentgenology. 2009 Jan;192(1):216.

6. Smith-Bindman R, Miglioretti DL, Rosenberg R, Reid RJ, Taplin SH, Geller BM, et al. Physician workload in mammography. American Journal of Roentgenology. 2008;190(2):526-32.

How come the time dedicated to reading diagnostic mammograms (diagnostic focus) was associated with gist experts in Table 2 when it was an unsignificant factor in Table 3, or do I read it wrong?

Thank you for the comment. That is an interesting observation. There are a few explanations for this:

(1) Due to the small sample size the correlation coefficients for the diagnostic focus did not yield a significant p-value, although it is above 0.2 and it is still more than most of the other characteristics. Larger sample size could lead to a significant p-value. This is added to the Discussion. 

(2) Also, the correlation coefficients were calculated for each pairwise comparison while categorizing readers as “Gist Experts” and “Others” were done based on the average of the AUCs from three pair-wise comparisons. As stated, the gist is a noisy signal, and averaging these three AUC values could reduce the noise and produce a performance metric more reflective of the user’s ability in detecting the gist signal. This explanation is added to the Result section. 

(3) for calculating the correlation coefficients, both gist experts and non-experts were pooled together. As an example, the scatter plot presented in Figure 1 shows the association between the AUC values for Cancer vs Normal categorization and the diagnostic focus. The red dots showed the “Gist Experts” while the blue dots showed “Other” radiologists. Within the “Other” group, a non-significant negative correlation (r=-0.21, p=0.30) and within the “Gist Experts”, a non-significant positive correlation (r=0.16, p=0.62) was noted. Figure 2 presents the boxplots for each category. As shown, the “Other” group included several radiologists, who stated that their diagnostic focus is less than 2.5% (about half of them). Even though the data points corresponding to these people reduced the correlation coefficient, the association of the diagnostic factor with being a “Gist Expert” was captured by the non-parametric test, i.e., Mann-Whitney U-test (p=0.004). 

Figure 1- A scatter plot showing the AUC values for “Cancer” vs “Normal” categorization against diagnostic focus.

Figure 2- Box plots showing the distribution of the AUC values for “Cancer” vs “Normal” categorization for “Gist Experts” and “Others”. 

A few minor comments:Abstract: The labels Chigh and Clow do not appear later in the text. (Why C? It refers to high and low gist signal, right?)

Thanks for raising this. Yes, that is correct. We omitted these abbreviations in the abstract. Previously they were only used in the abstract to ensure its word count is the required word limit.

Materials and Methods: 2-2 Mammograms. Explain CC.

Thanks for the comments. Fixed.

Results: Check “Prio_Vis” for consistency (incl. Discussion).

Thanks for raising this point. We fixed it throughout the manuscript.

---

## [Decision Letter · Decision Letter 1]

11 Jul 2022

PONE-D-21-31110R1Reliability of radiologists’ first impression when interpreting a screening mammogramPLOS ONE

Dear Dr. Ganodmkar:

Thank you for submitting your manuscript to PLOS ONE. After careful consideration, we feel that it has merit but does not fully meet PLOS ONE’s publication criteria as it currently stands. Therefore, we invite you to submit a revised version of the manuscript that addresses the points raised during the review process.

One reviewer accepted the work with no revisions, the other had some major revisions recommended.  Please address these in a revision. Please submit your revised manuscript by Aug 25 2022 11:59PM. If you will need more time than this to complete your revisions, please reply to this message or contact the journal office at plosone@plos.org. Please include the following items when submitting your revised manuscript:A rebuttal letter that responds to each point raised by the academic editor and reviewer(s). You should upload this letter as a separate file labeled 'Response to Reviewers'.A marked-up copy of your manuscript that highlights changes made to the original version. You should upload this as a separate file labeled 'Revised Manuscript with Track Changes'.An unmarked version of your revised paper without tracked changes. You should upload this as a separate file labeled 'Manuscript'.

We look forward to receiving your revised manuscript.

Kind regards,

Gayle E. Woloschak, PhD

Section Editor

PLOS ONE

Additional Editor Comments:

One reviewer accepted the work, the other had major revisions. Please address these in a revision.

Reviewers' comments:

Reviewer's Responses to Questions

**Comments to the Author**

1. If the authors have adequately addressed your comments raised in a previous round of review and you feel that this manuscript is now acceptable for publication, you may indicate that here to bypass the “Comments to the Author” section, enter your conflict of interest statement in the “Confidential to Editor” section, and submit your "Accept" recommendation.

Reviewer #2: All comments have been addressed

Reviewer #3: All comments have been addressed

2. Is the manuscript technically sound, and do the data support the conclusions?

Reviewer #2: Yes

Reviewer #3: Partly

3. Has the statistical analysis been performed appropriately and rigorously? 

Reviewer #2: Yes

Reviewer #3: Yes

4. Have the authors made all data underlying the findings in their manuscript fully available?

Reviewer #2: (No Response)

Reviewer #3: No

5. Is the manuscript presented in an intelligible fashion and written in standard English?

Reviewer #2: Yes

Reviewer #3: Yes

6. Review Comments to the Author

Reviewer #2: (No Response)

Reviewer #3: The manuscript is very difficult for reading. I suggest authors to perform substantial rewriting leaving only the most important information as main text. Additional analysis should be presented as supplemented material (for example, Tables 2, 4, Figures 2, 3, 4, 5, 6, 7, 8).

Authors should follow GRRAS guidelines (PMID: 21130355 DOI: 10.1016/j.jclinepi.2010.03.002)

METHODS

- Explain how the sample size was chosen. State the determined number of raters, subjects/objects, and replicate observations.

Authors didn’t provide any explanation for sample size for either participants or readers.

- Describe the sampling method.

This item was insufficiently reported: authors didn’t describe any method for avoiding selection bias of cases that were very suspicious of malignancy. This should be clearly stated and acknowledged as an important limitation.

- Describe the measurement/rating process (e.g. time interval between repeated measurements, availability of clinical information, blinding).

This item was insufficiently reported: authors didn’t describe whether the order of the images was the same between round 1 and round 2 and among the different readers.

- State whether measurements/ratings were conducted independently.

Authors didn’t state whether the reader was alone in the room while assessing the images.

- Describe the statistical analysis.

This item was not reported in the methods section. Authors should detail all the statistics that was used and the reason for doing that in the methods section.

Additionally, the authors’ criteria for creating a special class of readers, the “Gist Expert” is not adequate. Selecting a population based on the observed performance for making conclusions about their performance in the same dataset their were selected brings a lot of bias. This problem is similar to making conclusions about the diagnostic test accuracy of a test based only on the training dataset, excluding the need for a validation data-set. Authors should try to use baseline criteria to select a group, for example, the third working more hours per week reading mammograms, to evaluate whether reading more mammograms per week improve the readers ability.

The results section should be greatly reduced. The current presentation is very confusing. I strongly suggest authors to report this section using the following order.

RESULTS

- State the actual number of raters and subjects/objects which were included and the number of replicate observations which were conducted.

Authors could provide a better description of what was performed and whether there was any problem (for example, whether a reader didn’t return for the round 2).

- Describe the sample characteristics of raters and subjects (e.g. training, experience).

Authors might present Table 3, but the must add estimates of dispersion (min-max and p25-p75)

- Report estimates of reliability and agreement including measures of statistical uncertainty.

Authors should use a table to present the observed intra- (Round 1 vs Round 2) and inter-rater (Round 1 vs. Round 1) ICC and weighted Cohen’s kappa. Authors interpretation of ICC should also be based on the suggestion from GRRAS:

“Values of 0.60, 0.70, or 0.80 are often used as the minimum standards for reliability coefficients, but this may be only sufficient for group-level comparisons or research purposes [12,58,108]. For example, ICC values for a scale measuring pressure ulcer risk should be at least 0.90 or higher when applied in clinical practice [108]. If individual and important decisions are made on the basis of reliability estimates, values should be at least 0.90 [4] or 0.95 [109].”

Discussion

The current discussion section brings a lot of speculation regarding the effect of the readers’ first impression of their final evaluation of the mammogram, which should be completely removed as this was not assessed by the study.

Authors should focus on: key results, study limitations, and provide a careful interpretation of their results. My only interpretation based on the results of this study is that the intra-, and particularly, inter-observer reliability of the method is very poor and therefore this should not be used in clinical practice.

7. PLOS authors have the option to publish the peer review history of their article (what does this mean?). If published, this will include your full peer review and any attached files.

Reviewer #2: No

Reviewer #3: No

---

## [Author Response · Author response to Decision Letter 1]

22 Sep 2022

A word document containing the "Point by point responses to the comments" has been attached. The content of the file has been also provided below. Kindly please refer to the word document to see Table Res. 1 in the correct format.

 The manuscript is very difficult for reading. I suggest authors to perform substantial rewriting leaving only the most important information as main text. Additional analysis should be presented as supplemented material (for example, Tables 2, 4, Figures 2, 3, 4, 5, 6, 7, 8).

Authors should follow GRRAS guidelines (PMID: 21130355 DOI: 10.1016/j.jclinepi.2010.03.002)

Thank you very much for the comment. We believe addressing the comment considerably improved the presentation of our manuscript. Briefly, we amended the introduction and stated that the current study has four distinct aims. One of these aims (Aim 2-2) was to explore the test-retest reliability of the gist signal. To realize this aim we ensured that the GRRAS guideline is followed. Table Res. 1 shows the criteria in the GRRAS guidelines and how we addressed the criteria. 

While aim (2-2) investigates how reliable the gist signal is, aim (1) explores if the overall performance of a reader across all cases in two rounds (intra-radiologist variability in the performance) was associated. If the gist signal is generally present across the abnormal cases but picked up in a stochastic manner by an observer, the observer could detect the abnormality signal in one round but could miss it in another round. In such a scenario, an observer’s overall performance could be almost similar in two rounds, however, each time they detect the gist of the abnormal in different cases. This translates into low intra-radiologist variability in the overall performance but poor test-retest reliability of the gist signal. As our study showed that the intra-observer reliability is poor to modest, we also explored the magnitude of intra-observer variation in overall performance. Our data suggested that, when the gist signal is strong enough, i.e., with the cancer cases, the performance level in the first round of the gist experiment remained almost consistent in the second round. We also explored if we could predict which radiologists have a superior gist capability based on their characteristics (aim 2-1). Finally, aims (3) and (4) focus on the cases-level analysis to provide recommendations about whether an observer should trust their initial impression when a very strong gist of the abnormal is noted on a mammogram or conversely in the absence of a sense of an abnormal gist. This is important from a clinical point-of-view as it is unknown whether, in these extreme occasions, an observer’s initial impression can be trusted. Our findings suggest that the gist of the abnormal, especially a strong one, should not be ignored, because when the gist is strong, the probability of a case being abnormal is significantly higher than the probability that it is normal. Conversely, in the absence of a sense of an abnormal gist, readers should keep searching because the absence does not reliably signal a normal case. 

Moreover, as suggested by the reviewer, some of the analyses, tables, and figures are now moved to Supplementary Materials. 

Table Res. 1- The criteria in the GRRAS guidelines (Italic) and how we addressed them.

Title and abstract 1. Identify in title or abstract that interrater/intrarater reliability or agreement was investigated

In the abstract, it is stated that the study explored the intra- and inter-observer reliability of the reader’s gist signal. 

Introduction 2. Name and describe the diagnostic or measurement device of interest explicitly

Stated.

 3. Specify the subject population of interest

Stated.

 4. Specify the rater population of interest (if applicable)

Stated.

 5. Describe what is already known about reliability and agreement and provide a rationale for the study (if applicable)

Stated (The reliability is not known. The current study aimed at addressing this gap in the literature).

Methods 6. Explain how the sample size was chosen. State the determined number of raters, subjects/objects, and replicate observations.

Provided in the Sample size calculation. 

 7. Describe the sampling method

Participants (raters): the details are provided in Section 2-3 (Experimental Design and Study Participants). 

Images (objects): the details are provided in Section 2-4 (Mammograms). Further explanation is provided in the Supplementary Analysis 1.

 8. Describe the measurement/rating process (e.g., time interval between repeated measurements, availability of clinical information, blinding)

The details are provided in Section 2-3 (Experimental Design and Study Participants) and Section 3-1 (“On average, readers conducted the second round of the readings 7±1 weeks after the first round”).

 9. State whether measurements/ratings were conducted independently

Stated in Section 2-3 (Experimental Design and Study Participants). 

 10. Describe the statistical analysis

Stated in the 2-5- Statistical Analysis (aim 2-2). 

Results 11. State the actual number of raters and subjects/objects that were included and the number of replicate observations that were conducted.

Stated. Thirty-nine radiologists agreed to participate in the study, and all completed both rounds (no drop-out). For each reader in each round, 160 gist scores corresponding to 160 images were available. 

 12. Describe the sample characteristics of raters and subjects (e.g., training, experience)

Stated (Table 2 and 3). The raw data is available on Kaggle. 

 13. Report estimates of reliability and agreement including measures of statistical uncertainty

Stated. For the mean values the confidence interval and for the median values, min, max, fist and third quartiles were provided. 

Discussion 14. Discuss the practical relevance of results.

Stated. In summary, gist signal is not sufficiently reliable. However, our findings suggest that a strong gist of the abnormal, should not be ignored, because when the gist is strong, the probability of a case being abnormal is significantly higher than the probability that it is normal. Conversely, in the absence of a sense of an abnormal gist, readers should keep searching because the absence does not reliably signal a normal case. 

Auxiliary material 15. Provide detailed results if possible 

In the supplementary materials and the Kaggle all values and raw data are available. 

METHODS

- Explain how the sample size was chosen. State the determined number of raters, subjects/objects, and replicate observations.

Authors didn’t provide any explanation for sample size for either participants or readers.

A section is added to the Materials and Methods (Section 2-2) to describe the sample size calculation. 

- Describe the sampling method.

This item was insufficiently reported: authors didn’t describe any method for avoiding selection bias of cases that were very suspicious of malignancy. This should be clearly stated and acknowledged as an important limitation.

We added further explanation to Section 2-4, where the mammographic images were described. Moreover, in the Discussion, this is mentioned as a limitation of the study. 

The images were randomly retrieved from the breast cancer screening archive. Two experienced radiologists confirmed that the images had an acceptable quality in terms of breast positioning and acquisition parameters. “Normal” mammograms came from women who were normal at the time of examination and remained normal in the next round of screening. The experienced radiologists were asked to ensure none of the “Normal” cases contained benign findings or regions suspicious of malignancy. 

As stated in Discussion Section, did not include cases with benign findings in our “Normal” category. Moreover, we did not include the normal cases highly suspicious of malignancy. We only focused on two extremes, i.e., normal and abnormal, to study the gist signal. These two categories resulted in low and high gist scores. However, in real clinical practice, radiologists deal with benign findings and highly suspicious normal cases as well. As no benign cases were included here, we cannot exclude that a benign finding might evoke a gist signal as well. This could limit the usefulness of the gist signal for breast cancer detection in real clinical practice.

- Describe the measurement/rating process (e.g. time interval between repeated measurements, availability of clinical information, blinding).

This item was insufficiently reported: authors didn’t describe whether the order of the images was the same between round 1 and round 2 and among the different readers.

The time interval between repeated measurements: This is now added to Section 2-3 (Experimental Design and Study Participants) and Section 3-1 (“On average, readers conducted the second round of the readings 7±1 weeks after the first round.”).

Image Order: Images were presented in random order. Now stated in Section 2-3 (Experimental Design and Study Participants). In each session, the images were shuffled, and hence, the order of presentation of images was randomly assigned to each round and each observer.

- State whether measurements/ratings were conducted independently.

Authors didn’t state whether the reader was alone in the room while assessing the images.

Now stated in Section 2-3 (Experimental Design and Study Participants). They were alone in the room while assessing the images. 

- Describe the statistical analysis.

This item was not reported in the methods section. Authors should detail all the statistics that was used and the reason for doing that in the methods section.

A new section (Section 2-5, Statistical Analysis) is now added to the current revised version of the manuscript to address this comment. 

Additionally, the authors’ criteria for creating a special class of readers, the “Gist Expert” is not adequate. Selecting a population based on the observed performance for making conclusions about their performance in the same dataset their were selected brings a lot of bias. This problem is similar to making conclusions about the diagnostic test accuracy of a test based only on the training dataset, excluding the need for a validation data-set. Authors should try to use baseline criteria to select a group, for example, the third working more hours per week reading mammograms, to evaluate whether reading more mammograms per week improve the readers ability.

Thank you very much for the comment. In the current study, readers were also asked to complete a comprehensive questionnaire about their workload, practice type, experience levels, age, and gender, and. As shown in Table 2, the Pearson correlation coefficient was used to explore the association between the reader’s performance and the reader characteristics, which were collected as continuous/ordinal variables, including age, number of hours per week currently spent in reading mammograms, number of screening cases you read per week (screening interpretative volume), number of years certified as a BreastScreen reader, percentage of your time is dedicated to reading breast images, percentage of the time dedicated to reading the diagnostic mammograms, duration of fellowship training, number of years since fellowship training, number of years registered as a reader, number of years reading mammograms, number of breast biopsy examinations performed in the last 12 months, how often participate in a Multi-Disciplinary Meeting (MDT) in a month, and how often correlate/review radiology-pathology findings for biopsy cases in a month. For the dichotomous categorical variables (i.e., gender, whether they are radiologists or breast physicians, whether they are a screen reader, whether they are fellowship-trained, whether affiliated with a university or educational institutes, and whether they work full-time), Mann-Whitney U-test was used to investigate if the reader’s performance varied across two categories. However, none of the reader characteristics exhibited a significant relationship with the readers’ performance. Therefore, superiority in detecting the gist of the abnormal is an innate reader feature and at present, this capability can be only identified by measuring the performance of the reader on the gist task. Analysis of the overall performance of the readers suggests that, when the gist signal is strong enough, i.e., with the cancer cases, the performance level in the first round of the gist experiment remained almost consistent in the second round. In other words, readers who were “Gist Experts” based on their performance in the first round, were considered “Gist Experts” based on their performance in the second round as well. Therefore, although none of the reader characteristics were predictors of the gist performance, those with the capability of perceiving visible abnormalities based on their first impression, consistently outperformed others in the gist experiment.

The results section should be greatly reduced. The current presentation is very confusing. I strongly suggest authors to report this section using the following order. 

As per the reviewer’s suggestion, some of the results were moved to the supplementary materials. We also amended the manuscript to ensure that in all three sections of the manuscript (Materials and Methods, Results, and Discussion), the order of presented materials is in concordance with the order of study aims in the Introduction Section. 

RESULTS

- State the actual number of raters and subjects/objects which were included and the number of replicate observations which were conducted.

Authors could provide a better description of what was performed and whether there was any problem (for example, whether a reader didn’t return for the round 2).

None of the readers dropped out of the study. This information is now stated in Section 3-1. We also provided a further description of what was performed in Section 2-3 (Experimental Design and Study Participants).

- Describe the sample characteristics of raters and subjects (e.g. training, experience).

Authors might present Table 3, but the must add estimates of dispersion (min-max and p25-p75)

Thank you very much for raising this. In the revised version, Table 3 is updated as suggested. 

- Report estimates of reliability and agreement including measures of statistical uncertainty.

Authors should use a table to present the observed intra- (Round 1 vs Round 2) and inter-rater (Round 1 vs. Round 1) ICC and weighted Cohen’s kappa. Authors interpretation of ICC should also be based on the suggestion from GRRAS:“Values of 0.60, 0.70, or 0.80 are often used as the minimum standards for reliability coefficients, but this may be only sufficient for group-level comparisons or research purposes [12,58,108]. For example, ICC values for a scale measuring pressure ulcer risk should be at least 0.90 or higher when applied in clinical practice [108]. If individual and important decisions are made on the basis of reliability estimates, values should be at least 0.90 [4] or 0.95 [109].”

We updated Section 3-3 (The intra- and inter-reader reliability of gist responses) and provided a table showing all ICC and kappa values. To report the statistical uncertainty, either confidence interval or min-max and 25th-75th percentiles (in case of Median values) were provided. The ICC values were interpreted as per the reviewer’s suggestion.

Discussion

The current discussion section brings a lot of speculation regarding the effect of the readers’ first impression of their final evaluation of the mammogram, which should be completely removed as this was not assessed by the study. Authors should focus on: key results, study limitations, and provide a careful interpretation of their results. My only interpretation based on the results of this study is that the intra-, and particularly, inter-observer reliability of the method is very poor and therefore this should not be used in clinical practice.

Thank you very much for the comment. As suggested, we rewrote the Discussion section. To make it easier to follow, we outlined the key findings in an order similar to the study questions, presented in the Introduction. 

As noted by the reviewer, the current study shows that detecting this signal is noisy, with relatively low levels of inter- and intra-reader agreement. Despite the variations across individual readers, a near-perfect agreement in the gist scores for the average reader (the ICC of 0.96 (CI: 0.95-0.97) when comparing the average abnormality ratings from all readers) across the two rounds suggests that the strength of the gist signal is an unvarying innate property of the image, which may or not be perceived by a reader reliably. This opens up the opportunity for radiomic or other approaches to exploit this important, inherent image feature. Moreover, our case-level analysis (aim 4) showed a strong gist of the abnormal should not be ignored, because when the gist is strong, the probability of a case being abnormal is significantly higher than the probability that it is normal. Conversely, in the absence of a sense of an abnormal gist, readers should keep searching because the absence does not reliably signal a normal case. This supports the importance of “discovery scanning”, i.e., coarse screening to detect potential targets before terminating the visual search.

---

## [Decision Letter · Decision Letter 2]

22 Dec 2022

PONE-D-21-31110R2Reliability of radiologists’ first impression when interpreting a screening mammogramPLOS ONE

Dear Dr. Ganodmkar,

Thank you for submitting your manuscript to PLOS ONE. After careful consideration, we feel that it has merit but does not fully meet PLOS ONE’s publication criteria as it currently stands. Therefore, we invite you to submit a revised version of the manuscript that addresses the points raised during the review process. The paper was previously accepted by one reviewer, but this other reviewer has asked for major revisions.  Please address the concerns in a revised work.

We look forward to receiving your revised manuscript.

Kind regards,

Gayle E. Woloschak, PhD

Section Editor

PLOS ONE

Additional Editor Comments :

One reviewer had previously accepted the work. The other reviewer has again asked for major revisions. Please attempt to address those concerns in this revision.

Reviewers' comments:

Reviewer's Responses to Questions

**Comments to the Author**

1. If the authors have adequately addressed your comments raised in a previous round of review and you feel that this manuscript is now acceptable for publication, you may indicate that here to bypass the “Comments to the Author” section, enter your conflict of interest statement in the “Confidential to Editor” section, and submit your "Accept" recommendation.

Reviewer #3: (No Response)

2. Is the manuscript technically sound, and do the data support the conclusions?

Reviewer #3: Partly

3. Has the statistical analysis been performed appropriately and rigorously? 

Reviewer #3: Yes

4. Have the authors made all data underlying the findings in their manuscript fully available?

Reviewer #3: Yes

5. Is the manuscript presented in an intelligible fashion and written in standard English?

Reviewer #3: Yes

6. Review Comments to the Author

Reviewer #3: Authors need to change the abstract and discussion.

The observed intra-observer and, particularly, the inter-observer reprodiciblity (wehghted kappa = 0.3, ICC = 0.3) analyses have clearly shown that the method is not reliable and shouldn't be used in clinical practice.

Everything else in the manuscript is only disctraction and authors should focus on their main findings.

7. PLOS authors have the option to publish the peer review history of their article (what does this mean?). If published, this will include your full peer review and any attached files.

Reviewer #3: No

---

## [Author Response · Author response to Decision Letter 2]

7 Feb 2023

We would like to thank the editor and the reviewer for their valuable comments. The changes, in response to the comment, are highlighted in the manuscript. 

 Also, all de-identified collected data from radiologists, used in this study will be available through the following Kaggle dataset link:

https://www.kaggle.com/datasets/zibaga/gist-of-the-abnormal-on-screening-mammograms

PCB, JMW, SJL, KKE, DX, and ZG was funded by National Health and Medical Research Council (grant number APP1162872). Also, PCB, JMW, SJL, and KKE was funded by National Breast Cancer Foundation (IIRS-18-089). The funders had no role in study design, data collection and analysis, decision to publish, or preparation of the manuscript. We can confirm that we understand the above-mentioned statements will be included as “Funding Statement” in the manuscript. 

The authors have declared that no competing interests exist.

Point by point responses to the comments:

Reviewer #3:

Authors need to change the abstract and discussion.

The observed intra-observer and, particularly, the inter-observer reproducibility (wehghted kappa = 0.3, ICC = 0.3) analyses have clearly shown that the method is not reliable and shouldn't be used in clinical practice. Everything else in the manuscript is only distraction and authors should focus on their main findings.

Thank you very much for the comment. We rewrote the abstract and most of the Discussion to address this comment. In the revised version of the abstract, we solely explain the results indicating the test-retest reliability of the gist signal to ensure that the main message and finding of the paper are clear to the readers. We also amended the Discussion and mainly focused on discussing the findings concerning the intra- and inter-observer reliability of the signal. We ensured that it is clearly stated in both abstract and discussion that the gist signal has a poor test-retest reliability and hence should not be used in clinical practice.

---

## [Decision Letter · Decision Letter 3]

10 Feb 2023

PONE-D-21-31110R3Reliability of radiologists’ first impression when interpreting a screening mammogramPLOS ONE

Dear Dr. Ganodmkar,

Thank you for submitting your manuscript to PLOS ONE. After careful consideration, we feel that it has merit but does not fully meet PLOS ONE’s publication criteria as it currently stands. Therefore, we invite you to submit a revised version of the manuscript that addresses the points raised during the review process.

We look forward to receiving your revised manuscript.

Kind regards,

Gayle E. Woloschak, PhD

Section Editor

PLOS ONE

Journal Requirements:

Additional Editor Comments:

One reviewer has suggested one addition revision. Please address this.

Reviewers' comments:

Reviewer's Responses to Questions

**Comments to the Author**

1. If the authors have adequately addressed your comments raised in a previous round of review and you feel that this manuscript is now acceptable for publication, you may indicate that here to bypass the “Comments to the Author” section, enter your conflict of interest statement in the “Confidential to Editor” section, and submit your "Accept" recommendation.

Reviewer #3: All comments have been addressed

2. Is the manuscript technically sound, and do the data support the conclusions?

Reviewer #3: Partly

3. Has the statistical analysis been performed appropriately and rigorously? 

Reviewer #3: Yes

4. Have the authors made all data underlying the findings in their manuscript fully available?

Reviewer #3: Yes

5. Is the manuscript presented in an intelligible fashion and written in standard English?

Reviewer #3: Yes

6. Review Comments to the Author

Reviewer #3: The last two sentences of the discussion section are speculative and should be removed:

"Despite the variations across individual readers, a near-perfect agreement in the gist

scores for the average reader across the two experiments suggests that the strength of the gist signal is

an unvarying innate property of the image, which may or not be perceived by a reader reliably. This

opens up the opportunity for radiomic [36] or other approaches to exploit this important, inherent image

feature."

7. PLOS authors have the option to publish the peer review history of their article (what does this mean?). If published, this will include your full peer review and any attached files.

Reviewer #3: No

---

## [Author Response · Author response to Decision Letter 3]

27 Mar 2023

We would like to thank the editor and the reviewer for their valuable comments. The changes, in response to the comment, are highlighted in the manuscript. 

 Also, all de-identified collected data from radiologists, used in this study will be available through the following Kaggle dataset link:

https://www.kaggle.com/datasets/zibaga/gist-of-the-abnormal-on-screening-mammograms

PCB, JMW, SJL, KKE, DX, and ZG was funded by National Health and Medical Research Council (grant number APP1162872). Also, PCB, JMW, SJL, and KKE was funded by National Breast Cancer Foundation (IIRS-18-089). The funders had no role in study design, data collection and analysis, decision to publish, or preparation of the manuscript. We can confirm that we understand the above-mentioned statements will be included as “Funding Statement” in the manuscript. 

The authors have declared that no competing interests exist.

Point by point responses to the comments:

Reviewer #3:

The last two sentences of the discussion section are speculative and should be removed:

"Despite the variations across individual readers, a near-perfect agreement in the gist scores for the average reader across the two experiments suggests that the strength of the gist signal is an unvarying innate property of the image, which may or not be perceived by a reader reliably. This opens up the opportunity for radiomic [36] or other approaches to exploit this important, inherent image feature.".

Thank you very much for the comment. As suggested, the last two sentences were omitted in the revised manuscript.

---

## [Decision Letter · Decision Letter 4]

5 Apr 2023

Reliability of radiologists’ first impression when interpreting a screening mammogram

PONE-D-21-31110R4

Dear Dr. Ganodmkar,

We’re pleased to inform you that your manuscript has been judged scientifically suitable for publication and will be formally accepted for publication once it meets all outstanding technical requirements.

Kind regards,

Gayle E. Woloschak, PhD

Section Editor

PLOS ONE

Additional Editor Comments (optional):

Thank you for addressing the concerns of the reviewers.

Reviewers' comments:

Reviewer's Responses to Questions

**Comments to the Author**

1. If the authors have adequately addressed your comments raised in a previous round of review and you feel that this manuscript is now acceptable for publication, you may indicate that here to bypass the “Comments to the Author” section, enter your conflict of interest statement in the “Confidential to Editor” section, and submit your "Accept" recommendation.

Reviewer #3: All comments have been addressed

2. Is the manuscript technically sound, and do the data support the conclusions?

Reviewer #3: Yes

3. Has the statistical analysis been performed appropriately and rigorously? 

Reviewer #3: Yes

4. Have the authors made all data underlying the findings in their manuscript fully available?

Reviewer #3: Yes

5. Is the manuscript presented in an intelligible fashion and written in standard English?

Reviewer #3: Yes

6. Review Comments to the Author

Reviewer #3: The authors have properly addressed all the previous comments.

I don't have any further comments on this manuscript.

7. PLOS authors have the option to publish the peer review history of their article (what does this mean?). If published, this will include your full peer review and any attached files.

Reviewer #3: No

---

## [Editor Report · Acceptance letter]

11 Apr 2023

PONE-D-21-31110R4 

Reliability of radiologists’ first impression when interpreting a screening mammogram 

Dear Dr. Gandomkar:

I'm pleased to inform you that your manuscript has been deemed suitable for publication in PLOS ONE. Congratulations! Your manuscript is now with our production department. 

Kind regards, 

on behalf of

Dr. Gayle E. Woloschak 

Section Editor

PLOS ONE